# Multidimensional Relationships of Starch Digestibility with Physicochemical, Pasting and Textural Properties of 30 Rice Varieties

Liqin Hu [1], Jialin Cao [1], Yu Liu [1], Zhengwu Xiao [1], Mingyu Zhang [1], Jiana Chen [1], Fangbo Cao [1], Anas Iqbal [2], Salah Fatouh Abou-Elwafa [3] and Min Huang [1],*

[1]  Rice and Product Ecophysiology, Key Laboratory of Ministry of Education for Crop Physiology and Molecular Biology, Hunan Agricultural University, Changsha 410128, China; hnliqin1003@126.com (L.H.); caojl9595@163.com (J.C.); liuyudzhanghaol@163.com (Y.L.); xiaozhengwx@163.com (Z.X.); mingyuzhang@163.com (M.Z.); jianachen@hunau.edu.cn (J.C.); fangbocao@hunau.edu.cn (F.C.)
[2]  Key Laboratory of Crop Cultivation and Farming Systems, College of Agriculture, Guangxi University, Nanning 530004, China; anasiqbal@gxu.edu.cn
[3]  Agronomy Department, Faculty of Agriculture, Assiut University, Assiut 71526, Egypt; elwafa75@aun.edu.eg
*   Correspondence: mhuang@hunau.edu.cn

**Abstract:** Consuming rice with low starch digestibility is beneficial for reducing the risk of diabetes. Several factors have been shown to influence starch digestibility, but the combined effects of these factors on starch digestibility have not been studied. We assessed multidimensional relationships between the glucose production rate (GPR) of cooked rice with 16 indexes, including physicochemical, pasting and textural properties in 30 rice varieties. The stepwise multiple regression analysis showed that amylose content (AC), gel consistency (GC) and pasting temperature (PT) were closely related to GPR. This relationship could be described by the equation: GPR = $-0.080$ AC + 0.008 GC + 0.034 PT + 0.720, with a determination coefficient of 0.84. The variation partitioning analysis further indicated that AC, GC and PT independently explained 36%, 5% and 4% of the GPR variation, respectively. The interaction of AC and GC explained 46% of the variation in GPR. This study identifies the key indexes (AC, GC and PT) affecting starch digestibility and quantifies contributions of these indexes to the variation in GPR. The finding of our study provides useful information for breeding and selecting rice varieties with low GPR.

**Keywords:** rice; starch digestibility; amylose content; gel consistency; pasting temperature

## 1. Introduction

Rice is the food ingredient for two-thirds of the Chinese population [1], and its main component is starch, accounting for nearly 90% of its dry matter mass [2]. The digestibility of starch is an important aspect for health-conscious rice consumers, diabetics in particular [3]. In previous studies, it has been reported that higher rice consumption is more strongly associated with an increased risk of type 2 diabetes, especially in Asian populations such as Chinese and Japanese [4,5]. According to the *Ninth Edition of the International Diabetes Federation Statistics (2019)* [6], China had the highest number of diabetics globally in 2019 (116 million), and this trend is predicted to remain so in 2030 (140 million) and in 2045 (147 million). Diabetes is prevalent in China, which increases consumer demand for rice with low starch digestibility [7]. At present, it has been found that starch digestibility varies largely among different rice varieties, with the glycemic index (GI) ranging from 19 to 116 [8]. The diversity of GI provides an opportunity to select new varieties with a desirable postprandial glycemic response.

Several studies have been conducted to assess relationships of the starch digestibility of cooked rice with rice quality characteristics. For example, Fitzgerald et al. [9] observed a strong correlation ($r^2 = 0.73$) between the in vitro GI and amylose content (AC) across

235 diverse genotypes. Protein can form a complex with starch to decrease the glycemic response by forming a physical barrier [10,11]. Pasting properties also affect starch digestibility. For instance, viscosity breakdown is positively correlated with rapidly digestible starch content, whereas pasting temperature is positively correlated with resistant starch content [12,13]. Texture properties such as firmness were related to the estimated GI [14]. The relationships between starch digestibility and physicochemical, pasting and textural parameters are usually described by simple correlation analysis. Simple correlation is only used to investigate the relationship between two variables [15]. However, the multidimensional relationships between starch digestibility and various properties of rice have not been well studied.

In this study, we analyzed the multidimensional relationships of starch digestibility with physicochemical, pasting and textural properties of 30 rice varieties with the multiple stepwise regression analysis and variation partitioning analysis. The study aimed to: (i) identify the key indexes affecting starch digestibility; and (ii) quantify contributions of the key indexes affecting starch digestibility.

## 2. Samples and Methods

### 2.1. Preparation of Rice Sample

Thirty representative materials were selected from the College of Agronomy, Hunan Agricultural University, based on different amylose content. After rice harvest in 2019, grain samples were stored at room temperature for at least three months to obtain stable moisture of 13%, and then were dehulled and milled using a milled rice testing machine (JGMJ8098, Shanghai Jiading Cereals and Oils Instrument Co., Ltd., Shanghai, China). Rice flour (100 mesh) and cooked rice were prepared according to the method described by Huang et al. [16].

### 2.2. Measurements

#### 2.2.1. Physicochemical Parameters

The physicochemical parameters of rice flour, including total starch (TS), amylose content (AC), protein content (PC) and gel consistency (GC) were measured in the grain samples of the 30 studied rice varieties. All analyses were conducted in triplicate and results were presented as mean on a dry matter basis (DM). TS was measured using an auto digital polarimeter (P850 Pro; Jinan Hanon Instruments Co., Ltd., Jinan, China). AC was determined by iodine colorimetry according to the procedure of Juliano [17]. Standard curves were formed using standard samples containing 0.4%, 10.3%, 16.6% and 26.6% AC, provided by the China Rice Research Institute (CNRRI). PC was determined according to the method described by Huang et al. [18]. The quantified nitrogen of the rice flour was converted to protein content by the factor of 5.95. GC was measured using the method of Capampang et al. [19]. Those values are expressed on a dry-weight basis. Moisture content was determined by pre-weighing, then drying grain samples at 105 °C to a constant weight.

#### 2.2.2. Pasting Parameters

Pasting parameters of milled rice flour were measured using a Rapid Visco Analyzer (RVA-Super 4, Newport Scientific Pty Ltd., Warriewood, Australia) according to Zhu et al. [20] with minor modifications. In brief, $3.00 \pm 0.01$ g of rice flour and 25 mL distilled water were mixed in an aluminum canister. RVA analysis was carried out according to the following program: holding at 50 °C for 1 min, heating from 50 °C to 95 °C at 12 °C/min, holding at 95 °C for 2.5 min and cooling from 95 °C to 50 °C at 11.69 °C/min rate, remaining at 50 °C for 1.4 min. All analyses were conducted in triplicate and results were presented as mean. The following pasting parameters were determined: peak viscosity (PV), trough viscosity (TV), breakdown viscosity (BV), final viscosity (FV), setback viscosity (SV), peak time (PKT, min) and pasting temperature (PT, °C). All the viscosity parameters were expressed in cP.

### 2.2.3. Textural Parameters

Texture profile analysis (TPA) was conducted with a texture analyzer (Rapid TA+, Tengba Co., Shanghai, China). Three rice grains were selected from almost the same aluminum boxes after removing the first layer of cooked rice, and were placed on the base plate. The TPA mode was employed to analyze texture parameters of the cooked rice grains. A cylinder-type plunger (10 mm diameter) compressed the rice grains at a crosshead speed of 0.5 mm s$^{-1}$ at a strain of 50%. The procedure was repeated at least six times for each sample. Results were expressed as mean. Parameters recorded from TPA curves include hardness (HRD, g), springiness (SPR), cohesiveness (COH), chewiness (CHW, g) and resilience (RES).

### 2.2.4. Starch Digestion Parameters of Cooked Rice

Starch digestion parameters of the cooked rice were determined according to the method described by Huang et al. (2021) [16]. In brief, cooked rice samples were placed into the base of the simulated masticator (Zyliss, Zurich, Switzerland). Rice samples were chopped 20 times to fine particles of approximately 2–3 mm in size. An amount of 100 mg of the chewed rice grains was placed into a sample cup of the in vitro digestion simulator machine (Nutri Scan GI20, National Instruments, Australia) that contains a stirrer bar and was kept in a heating block to maintain its temperature at 37 °C. A three-enzyme solution ($\alpha$-amylase, pepsin, a mixture of pancreatin and amyloglucosidase) was added to each sample cup in a specific sequence along with buffer solutions (0.02 M NaOH and 0.2 M sodium acetate). After samples were digested as occurs inside the human system, Glucose Analyser (GM9, Analox, UK) was employed to measure the amount of glucose released from these digested samples at different time points (15, 60, 120, 180, 240 and 300 min). All analyses were conducted in triplicate and results were presented as mean. The detailed detection procedure is shown in Figure 1.

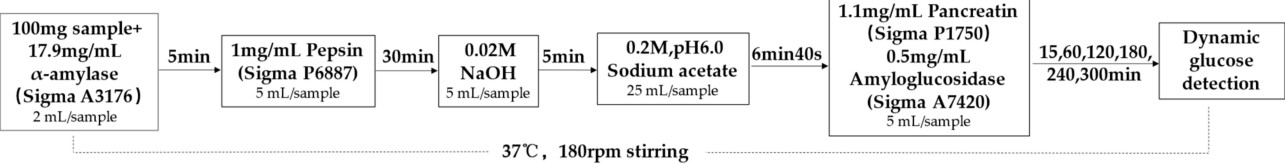

**Figure 1.** The flow chart of digestion method of GI20 instrument.

A non-linear model was applied to describe the kinetics of starch digestibility [21]. The first-order equation has the form:

$$y = a\,[1 - \exp(-bx)] \tag{1}$$

where x is the digestion time (min), y is the amount of produced glucose (g/100 g sample) at x time, *a* is the final amount of produced glucose after 300 min and *b* is the constant kinetic parameter. Parameters *a* and *b* were estimated using the Curve Expert software 1.4 (Hyams Development, Chattanooga, TN, USA) based on the data obtained from the in vitro digestion procedure. Furthermore, the starch digestibility parameters, including total glucose production (TGP), active digestion duration (ADD) and glucose production rate (GPR), were calculated with y at 95% of *a* (0.95*a*) based on Equation (1):

$$TGP = 0.95a \tag{2}$$

$$ADD = \ln(0.05)/-b \tag{3}$$

$$GPR = TGP/ADD \tag{4}$$

where *a*, *b* are the parameters of Equation (1).

### 2.3. Statistical Analysis

The multiple stepwise regression analysis was performed to identify the multidimensional relationship of starch digestibility with physicochemical, pasting and textural properties of 30 rice varieties (SPSS software version 26.0, IBM, Armonk, NY, USA). Variation partitioning analysis was performed to measure the contributions of identified key factors to the GPR of cooked rice (R studio, Boston, MA, USA; version 1.4.1103) [22].

## 3. Results

### 3.1. Physicochemical, Pasting and Textural Parameters of the Sample

Significant differences were observed among the 30 rice varieties in most of the measured rice parameters (Table 1). Physicochemical parameters such as TS, PC, AC and GC of the 30 rice varieties ranged from 84.2 to 90.2 g/100 g, 5.33 to 10.6 g/100 g, 11.7 to 29.0 g/100 g and 27 to 98 mm, respectively. The RVA parameters, i.e., PV, TV, BV, FV, SV, PKT and PT ranged from 527 to 4070 cP, 156 to 3173 cP, 372 to 2228 cP, 367 to 4959 cP, −1288 to 1168 cP, 4.71 to 6.78 min and 72.1 to 90.8 °C, respectively. Similarly, textural parameters such as HRD, SPR, CHW, COH and RES ranged from 462 to 1442 g, 0.52 to 1.10, 109 to 726 g, 0.39 to 0.64 and 0.26 to 0.47, respectively. The variable coefficients of 12 out of 16 traits including SV, BV, CHW, GC, TV, HRD, AC, FV, PV, PC, RES and COH (in descending order) were more than 10%. The results showed great differences in quality traits among 30 rice varieties.

**Table 1.** Physicochemical, pasting and textural parameters of 30 rice varieties.

| Variety | Physicochemical Parameters [a] | | | | RVA Parameters [b] | | | | | | | Textural Parameters [c] | | | | |
|---|---|---|---|---|---|---|---|---|---|---|---|---|---|---|---|---|
| | TS (g/100 g) | PC (g/100 g) | AC (g/100 g) | GC (mm) | PV (cP) | TV (cP) | BV (cP) | FV (cP) | SV (cP) | PKT (min) | PT (°C) | HRD (g) | SPR | CHW (g) | COH | RES |
| Zhongzao 39 | 84.5 | 8.1 | 24.8 | 30 | 3244 | 2380 | 864 | 4282 | 1039 | 6.00 | 82.9 | 1388 | 0.76 | 657 | 0.62 | 0.46 |
| Luliangyou 996 | 85.2 | 9.6 | 27.1 | 27 | 3196 | 2640 | 556 | 4258 | 1062 | 6.13 | 82.9 | 1374 | 0.74 | 623 | 0.61 | 0.44 |
| Zhongjiazao 17 | 85.9 | 8.1 | 27.1 | 29 | 3236 | 2283 | 953 | 4320 | 1084 | 5.78 | 81.6 | 1442 | 0.78 | 715 | 0.64 | 0.47 |
| Zhuliangyou 729 | 87.3 | 7.4 | 26.7 | 27 | 3309 | 2594 | 716 | 4346 | 1037 | 6.05 | 82.6 | 1320 | 0.75 | 607 | 0.61 | 0.45 |
| Guangluai 4 | 86.5 | 10.1 | 28.4 | 38 | 3312 | 2433 | 879 | 4430 | 1118 | 5.75 | 81.0 | 1224 | 0.72 | 563 | 0.64 | 0.46 |
| Xiangzaoxian 24 | 84.5 | 10.1 | 26.3 | 36 | 3269 | 2604 | 665 | 4437 | 1168 | 6.22 | 81.6 | 933 | 0.69 | 362 | 0.56 | 0.40 |
| Yuezaoxian 17 | 84.2 | 10.0 | 27.1 | 54 | 3262 | 2462 | 800 | 4034 | 772 | 6.09 | 79.2 | 896 | 1.10 | 726 | 7.56 | 0.45 |
| e1703 | 87.0 | 8.3 | 29.0 | 95 | 4070 | 2908 | 1162 | 4959 | 889 | 6.07 | 80.8 | 946 | 0.66 | 370 | 0.57 | 0.39 |
| Xiangzaoxian 45 | 84.8 | 10.6 | 14.4 | 71 | 1577 | 1127 | 451 | 2213 | 635 | 6.09 | 90.8 | 1047 | 0.64 | 382 | 0.57 | 0.40 |
| Qiliangyou 2012 | 87.4 | 9.9 | 14.5 | 91 | 527 | 156 | 372 | 364 | −164 | 4.71 | 75.2 | 780 | 0.56 | 205 | 0.47 | 0.31 |
| Y-liangyou 900 | 90.2 | 6.9 | 12.2 | 96 | 4026 | 1799 | 2228 | 2738 | −1288 | 5.60 | 81.5 | 462 | 0.59 | 121 | 0.43 | 0.29 |
| Deyou 4727 | 89.8 | 7.3 | 13.2 | 92 | 3868 | 1828 | 2040 | 2793 | −1075 | 5.78 | 74.2 | 488 | 0.56 | 119 | 0.43 | 0.28 |
| Jingliangyou 1468 | 88.1 | 8.3 | 16.2 | 73 | 2859 | 1440 | 1419 | 2431 | −428 | 5.91 | 72.1 | 695 | 0.65 | 222 | 0.48 | 0.34 |
| Liangyoupei jiu | 87.2 | 8.2 | 24.8 | 82 | 2296 | 1373 | 923 | 2639 | 343 | 5.98 | 77.1 | 836 | 0.62 | 290 | 0.55 | 0.37 |
| Guiliangyou 2 | 89.4 | 7.1 | 26.9 | 98 | 3219 | 2093 | 1127 | 3577 | 358 | 5.78 | 80.7 | 940 | 0.65 | 344 | 0.56 | 0.40 |
| Y-liangyou 1 | 89.6 | 6.9 | 11.7 | 84 | 3799 | 1969 | 1829 | 2905 | −893 | 5.85 | 75.5 | 638 | 0.52 | 161 | 0.46 | 0.32 |
| Jinnongsimiao | 87.6 | 8.5 | 12.2 | 90 | 3750 | 2036 | 1714 | 2931 | −819 | 5.87 | 76.1 | 517 | 0.54 | 109 | 0.39 | 0.26 |
| Yuxiangyouzhan | 88.3 | 7.0 | 24.2 | 55 | 2863 | 1838 | 1025 | 3221 | 358 | 6.15 | 72.6 | 1144 | 0.69 | 474 | 0.58 | 0.42 |
| Zhenguiai | 85.9 | 9.0 | 26.8 | 33 | 3630 | 3037 | 594 | 4542 | 911 | 6.35 | 79.5 | 1347 | 0.74 | 629 | 0.62 | 0.46 |
| Guichao 2 | 86.2 | 8.1 | 28.2 | 50 | 3666 | 3173 | 493 | 4223 | 557 | 6.78 | 84.7 | 1110 | 0.69 | 487 | 0.62 | 0.46 |
| Meixiangzhan 2 | 88.6 | 6.5 | 18.7 | 66 | 3355 | 1786 | 1569 | 3253 | −101 | 5.67 | 83.6 | 840 | 0.72 | 339 | 0.55 | 0.40 |
| Xiangyaxiangzhen | 88.1 | 6.9 | 19.1 | 69 | 3823 | 1902 | 1921 | 3292 | −531 | 5.72 | 75.1 | 824 | 0.68 | 299 | 0.52 | 0.38 |
| Yuzhenxiang | 87.8 | 7.6 | 17.2 | 66 | 3593 | 1851 | 1742 | 3229 | −364 | 5.71 | 82.4 | 987 | 0.66 | 337 | 0.51 | 0.37 |
| Taiyou 871 | 88.7 | 6.5 | 17.0 | 55 | 3374 | 1753 | 1621 | 3239 | −135 | 5.69 | 85.3 | 893 | 0.66 | 316 | 0.52 | 0.38 |
| Xiangwanxian 17 | 88.3 | 7.3 | 17.9 | 66 | 3460 | 1798 | 1662 | 3292 | −168 | 5.72 | 84.6 | 985 | 0.69 | 355 | 0.52 | 0.38 |
| Wuyou 308 | 87.9 | 5.8 | 22.9 | 52 | 3437 | 2053 | 1384 | 3978 | 542 | 5.75 | 79.4 | 1104 | 0.68 | 463 | 0.62 | 0.46 |
| Jiyou 225 | 88.2 | 6.3 | 23.5 | 46 | 3651 | 2094 | 1557 | 3919 | 268 | 5.67 | 78.6 | 1165 | 0.71 | 508 | 0.59 | 0.44 |
| Tianyouhuazhan | 88.0 | 6.1 | 23.8 | 41 | 3558 | 1987 | 1571 | 3801 | 243 | 5.66 | 79.0 | 1020 | 0.70 | 413 | 0.57 | 0.43 |
| H-you 518 | 87.9 | 6.5 | 17.5 | 63 | 3517 | 1730 | 1787 | 3114 | −403 | 5.62 | 78.3 | 997 | 0.64 | 329 | 0.52 | 0.38 |
| Wufengyou T025 | 88.3 | 5.3 | 23.2 | 60 | 3609 | 2184 | 1425 | 4127 | 518 | 5.81 | 79.1 | 1060 | 0.72 | 484 | 0.62 | 0.46 |
| CV (%) [d] | 1.9 | 17.3 | 26.6 | 37.3 | 21.9 | 29.1 | 42.7 | 26.5 | 327.7 | 5.8 | 5.1 | 26.8 | 9.8 | 42.7 | 12.2 | 14.9 |

[a] TS, total starch; PC, protein content; AC, amylose content; GC, gel consistency. [b] PV, peak viscosity; TV, trough viscosity; BV, breakdown viscosity; FV, final viscosity; SV, setback viscosity; PKT, peak time; PT, pasting temperature. [c] HRD, hardness; SPR, springiness; COH, cohesiveness; CHW, chewiness; RES, resilience. [d] CV, coefficient of variation.

### 3.2. Starch Digestion Parameters of Cooked Rice

The kinetics profiles of starch digestion showed significant variations in the total glucose production (TGP), active starch digestion duration (ADD) and glucose production rate (GPR) among the thirty studied rice varieties (Table 2). The TGP of cooked rice ranged from

294 to 365 mg/g. The highest and lowest TGP values were produced from the Jiyou 225 and Yuezaoxian 17 varieties. The ADD ranged from 923 to 250 min, with the longest ADD resulting from the Zhuliangyou 729 variety and the shortest ADD resulting from the Xiangzaoxian 45 variety. The GPR ranged from 1.3 to 3.6 mg/g/min; Xiangzaoxian 45 and Zhenguiai exhibited the fastest and slowest GPR, respectively. The coefficients of variation for TGP, ADD and GPR were 5.08%, 26.9% and 27.8%, respectively. These results indicate that the main reason for the significant difference in GPR among rice varieties was the variation in ADD, not TGP.

**Table 2.** Starch digestion properties (GPR, glucose production rate; ADD, active digestion duration; and TGP, total glucose production) of cooked rice of 30 rice varieties.

| Variety | TGP (mg/g) | ADD (min) | GPR (mg/g/min) |
|---|---|---|---|
| Zhongzao 39 | 317 | 221 | 1.43 |
| Luliangyou 996 | 333 | 214 | 1.56 |
| Zhongjiazao 17 | 319 | 184 | 1.73 |
| Zhuliangyou 729 | 338 | 250 | 1.35 |
| Guangluai 4 | 325 | 209 | 1.56 |
| Xiangzaoxian 24 | 311 | 180 | 1.73 |
| Yuezaoxian 17 | 294 | 184 | 1.60 |
| e1703 | 304 | 158 | 1.92 |
| Xiangzaoxian 45 | 338 | 93 | 3.63 |
| Qiliangyou 2012 | 317 | 116 | 2.73 |
| Y-liangyou 900 | 359 | 126 | 2.85 |
| Deyou 4727 | 335 | 111 | 3.02 |
| Jingliangyou 1468 | 313 | 116 | 2.70 |
| Liangyoupei jiu | 320 | 143 | 2.24 |
| Guiliangyou 2 | 338 | 154 | 2.19 |
| Y-liangyou 1 | 318 | 111 | 2.86 |
| Jinnongsimiao | 327 | 112 | 2.92 |
| Yuxiangyouzhan | 341 | 169 | 2.02 |
| Zhenguiai | 314 | 234 | 1.34 |
| Guichao 2 | 312 | 191 | 1.63 |
| Meixiangzhan 2 | 319 | 137 | 2.33 |
| Xiangyaxiangzhen | 323 | 140 | 2.31 |
| Yuzhenxiang | 346 | 112 | 3.09 |
| Taiyou 871 | 353 | 110 | 3.21 |
| Xiangwanxian 17 | 345 | 135 | 2.56 |
| Wuyou 308 | 340 | 159 | 2.14 |
| Jiyou 225 | 365 | 170 | 2.15 |
| Tianyouhuazhan | 346 | 210 | 1.65 |
| H-you 518 | 332 | 127 | 2.61 |
| Wufengyou T025 | 346 | 149 | 2.32 |
| CV (%) [a] | 5.08 | 26.9 | 27.8 |

[a] CV, coefficient of variation.

### 3.3. Correlation between Digestion Properties, Physicochemical, Pasting and Textural Parameters

The independent variables were divided into three categories, i.e., physicochemical, pasting and textural parameters, and the dependent variable was GPR. GPR was subjected to stepwise regression analysis with these three sets of traits separately or together. The results are shown in Table 3. The findings revealed that the key physicochemical parameters that influence GPR were AC and GC. It showed a regression coefficient of 0.79, while AC and GPR had a regression determination coefficient of 0.76. The most important RVA indexes, i.e., SV, TV and PT, showed a regression coefficient of 0.76 with the GPR. The primary texture index, CHW, exhibited a regression coefficient of 0.57 with GPR. The multiple regression analysis results revealed that physicochemical and pasting parameters were the main factors influencing GPR, and the prediction accuracy of texture parameters for GPR was relatively low. In addition, regression analysis of GPR with all estimated indices re-

vealed that AC, GC and PT were the most important factors in the model for estimating the GPR of the cooked rice. The equation was GPR = −0.08 AC + 0.008 GC + 0.034 PT + 0.720, with an $R^2$ of 0.84 ($p < 0.01$).

**Table 3.** The stepwise multiple regression analysis of physicochemical, pasting and textural parameters to GPR.

| Independent Variable | Regression Equation [a] | $R^2$ |
|---|---|---|
| Physicochemical parameters | GPR = −0.095 AC + 4.29 | 0.76 ** |
|  | GPR = −0.08 AC + 0.007 GC + 3.55 | 0.79 ** |
| Pasting parameters | GPR = −0.001 SV − 0.0005 TV + 0.054 PT − 1.031 | 0.76 ** |
| Textural parameters | GPR = −0.003 CHW + 3.32 | 0.57 ** |
| All parameters | GPR = −0.08 AC + 0.008 GC + 0.034 PT + 0.720 | 0.84 ** |

[a] GPR, glucose production rate; AC, amylose content; GC, gel consistency; SV, setback viscosity; TV, trough viscosity; PT, pasting temperature; CHW, chewiness. ** denotes a significance at the $p \leq 0.01$ level.

*3.4. Relative Contributions of Amylose Content, Gel Consistency and Pasting Temperature to GPR*

According to the stepwise multiple regression analysis results, the multiple regression determination coefficients of the AC, GC and PT on the GPR were the highest, so we assessed the contribution of these three significant factors to the GPR according to variation partitioning analysis (Figure 2). AC, GC and PT explained 84% of the observed variations in GPR of the cooked rice ([a] + [b] + [c] + [d] + [e] + [f] + [g]). AC explained 36% of the GPR variations ([a]), which is the most of any single factor. GC and PT accounted for 5% ([b]) and 4% ([c]) of the GPR variation, respectively. AC and GC together explained 46% of the GPR variations ([d] + [g]), which was greater than the explained AC contribution alone. Therefore, GC was another important factor that affected GPR. The shared variations explained by PT with AC [f], PT with GC [e] and PT with AC and GC [g] were relatively low. PT had a weak contribution to GPR compared with AC and GC.

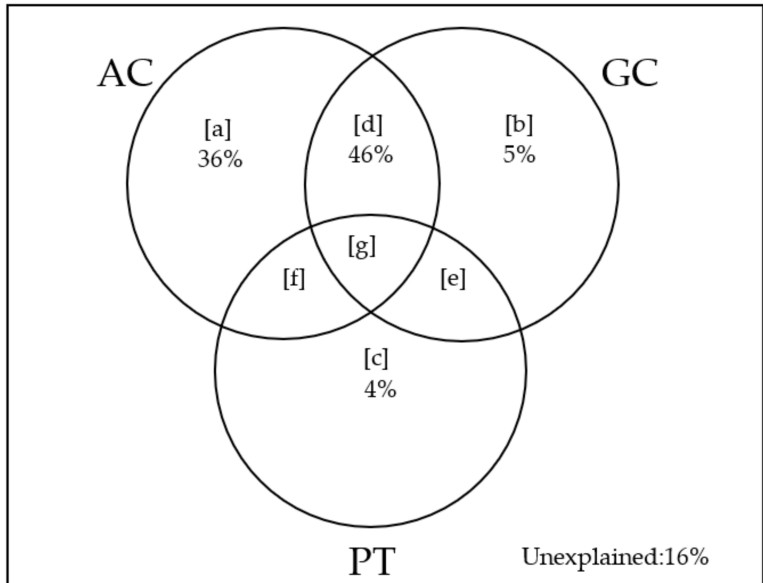

**Figure 2.** Variation partition analysis of the effects of AC, GC and PT on GPR. The variations in the GPR explained by the three factors are expressed in percentages. Amylose content (AC) [a], gel consistency (GC) [b], pasting temperature (PT) [c] and shared fractions [d], [e], [f], [g] are provided separately. Negative values are not shown.

## 4. Discussion

It has been debated whether AC can be used as a decisive factor in evaluating the starch digestibility of cooked rice. Studies have reported that AC is negatively correlated

with starch digestibility due to the fact that amylose contains a large number of intra- and intermolecular hydrogen bonds, and the microcrystalline structure limits the extent of starch swelling and gelatinization during heating, thus reducing starch digestibility [22,23]. However, it has also been shown that AC is not a very effective index for measuring the carbohydrate digestion properties of rice [24,25]. In our study, the multiple regression coefficients of determination ($R^2$) between GPR and AC were 0.77. This is consistent with Fitzgerald et al. [9], who found a correlation ($R^2$ = 0.73) between predicted GI and AC across 235 diverse samples. AC explained 36% of the GPR variations, which was the factor that contributed the most to the variation of GPR. Chung et al. (2011) [26] and Hu et al. (2004) [27] have reported the same conclusion.

In the present study, Xiangzaoxian 45 and Qiliangyou 2012; Liangyoupeijiu and Zhongzao 39; Guichao 2 and Guangluai 4 have similar amylose content, but the GPR is different. This agrees with the previous report that rice varieties with similar AC could have different digestibility, possibly due to their physicochemical (particularly gelatinization) properties [28]. Therefore, AC is the major factor in determining starch digestibility, but not the only one. Panlasigui et al. [29] had found that GI was negatively correlated with GC and significantly correlated with the alkaline diffusion value (ASK is negatively correlated with PT) through in vivo experiments. This study revealed that GC explained 5% of the variation in the GPR, but AC and GC interaction explained 46% of the variation in GPR through variation partitioning analysis. Therefore, GC can complement the AC in a breeding program for evaluating the starch digestibility of cooked rice, because GC reflects the combined effect of amylose content and the molecular properties of amylose and amylopectin [19]. PT just explained 4% of the GPR variation. Consequently, PT can affect starch digestibility but is not the main factor when compared to the AC and GC.

According to the formula, GPR = −0.08 AC + 0.008 GC + 0.034 PT + 0.720, it is obvious that varieties with low GPR had the characteristics of high AC, low GC and low PT values. It is generally accepted that AC is negatively correlated with GC and positively correlated with PT [26,30,31]. However, a few studies have reported that AC exhibits a negative correlation with both the GC and PT [32,33], so these special materials may be the varieties with low GPR, which needs further verification. AC, GC and PT are known in most varieties because they have been used to evaluate rice taste [34,35], so it would be high-throughput, convenient and inexpensive to screen varieties with lower GPR among the existing varieties. In addition, previous studies have indicated that the three traits that are highly correlated at the genetic level, such as Wx and SSII-3, play a major role in regulating AC, GC and PT. SBE3 and ISA affect GC and PT simultaneously, while SSIII-2, AGPlar, PUL and SSI are specific for AC; AGPiso for GC; and SSIV-2 for PT [32,36,37]. Therefore, breeders can increase AC and reduce GC and PT by regulating the starch metabolic pathway to select new varieties with low GPR, so as to meet the demand of people with diabetes.

## 5. Conclusions

This study identifies three key indexes (AC, GC and PT) affecting starch digestibility of cooked rice. These three indexes can explain more than 80% of the variation in GPR. The interaction of AC and GC has the highest contribution to the variation in GPR. High AC combined with low GC and PT can be used as new criteria for selecting and breeding rice varieties with low GPR.

**Author Contributions:** Conceptualization, M.H.; data curation, L.H., J.C. (Jiana Chen) and F.C.; formal analysis, L.H.; funding acquisition, M.H.; investigation, L.H., J.C. (Jialin Cao) and Y.L.; methodology, L.H., Z.X. and M.Z.; project administration, M.H.; writing—original draft, L.H.; writing—review and editing, A.I., S.F.A.-E. and M.H. All authors have read and agreed to the published version of the manuscript.

**Funding:** This research was funded by the National Key R&D Program of China: 2016YFD0300509.

**Institutional Review Board Statement:** Not applicable.

**Informed Consent Statement:** Not applicable.

**Data Availability Statement:** Data are contained within the article.

**Conflicts of Interest:** The authors declare no conflict of interest.

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
