# Peer review of "Multidimensional Relationships of Starch Digestibility with Physicochemical, Pasting and Textural Properties of 30 Rice Varieties"

_agronomy, doi:10.3390/agronomy12030720_

Round 1
Reviewer 1 Report
The manuscript reported the study of the correlations between 16 indexes,
(Including physicochemical, pasting, and textural properties) with the glucose production rate (GPR), an indicator of starch digestion properties, in 30 rice varieties, showing that amylose content (AC), gel consistency (GC) and pasting temperature (PT) were the key factors affecting GPR.
The focus of the work is to establish a convenient method to identify rice varieties with a low glycemic index. The authors provided extensive data and statistical analysis; the paper lacks the explanation and discussion of the significance of the parameters analyzed (GC, PT and so on). The authors did not discuss the dependence of key parameters (GC and PT) on the amylose content, well known in the literature. From the results, it looks like the only significant parameter valuable for the prediction of starch digestibility is amylose content as extensively reported in the literature. English is clear but some spelling errors are present. The introduction and the discussion are poor and do not allow to clearly perceive the value of their work; moreover, the sentence on starch related gene is ambiguous. the method proposed by the author is laborious and do not fit well with the need of a rapid screening.
Reviewer 2 Report
The ms. “Correlation 1 of starch digestion with physicochemical, pasting and textural properties of 30 rice varieties” presents original research regarding correlation between 16 indicators (total starch; protein content, amylose content, gel consistency, peak viscosity, trough viscosity, breakdown viscosity, final viscosity, setback viscosity, peak time, pasting temperature, hardness, springiness, cohesiveness, chewiness, resilience) with the glucose production rate (GPR) in 30 rice varieties.
In vivo experiments give the full view of starch digestibility as they take into account other aspects as well, but are time consuming and expensive. After consumption of food containing resistant starch, among other factors, the influence of fermentation in the large intestine by microorganisms positively affecting the host organism is observed, among others. The degree of fragmentation of the raw material and the cooking temperature are also important. The differences in the obtained glycemic index may be very large due to these factors.
Nevertheless authors , by creating the same conditions for the preparation of the raw material for different rice varieties, the authors of the presented work determined what can affect glucose production rate to the greatest extent, and they are high amylose content (AC), low gel consistency (GC) and low pasting temperature (PT). Based on the examination of these parameters, the digestibility of rice starch can be quickly determined.
I have no comments about the research methods used and the remaining chapters of the work.
Reviewer 3 Report
The authors of this manuscript have described the importance of their approach of studying an essential approach of studying starch digestibility in rice plants. In the whole this article is well-structured and contains a lot of necessary information about methods and progress of previous researchers in this field.
However there are several lexical and grammatical mistakes.
Point 1 (Line 13). You should replace the adverb “together” by “combined effects”.
Point 2 (Line 21). You should replace by “rice varieties with low GPR”.
Point 3 (Line 24). It will be better to write “food ingredient for two thirds…”
Point 4 (Lines 24, 25, 27, 29, 30, 33, 40, 44). You should insert the space before the references.
Point 5 (Line 26). You should correct the spelling “diabetics”.
Point 6 (Line 27). You should replace the part of sentence by “ In previous studies it has been reported…” to make the sentence grammatically correct.
Point 7 (Line 30). You should replace “has” by “had” because you write “in 2019” and it corresponds to Past Simple.
Point 8 (Line 42). You should correct the spelling “numerous”.
Point 9 (Line 45-46). You should replace “are required” by “is required” because the subject “development” is singular.
Point 10 (Line 49, 51, 53, 55). You should insert the space before the references.
Point 11 (Line 54). You should replace “was found” by “were found” because the subject “properties” is plural.
Point 12 (Line 55). You should replace “relationship” by “relationships” because the predicate “are described” is for plural subject.
Point 13 (Line 65). You should revise the spelling “glycemic”.
Point 14 (Line 68). You should replace “materials” by “samples”.
Point 15 (Line 79). You should correct the spelling “varieties”.
Point 16 (Line 83). You should replace the predicate by “were provided” yo make the sentence grammatically correct.
Point 17 (Line 94-95). You should replace your version by “holding at 50 ℃, holding at 95 ℃, cooling…, remaining…
Point 18 (Line 104). You should replace “were employed” by “was employed” because the subject “mode” is singular.
Point 19 (Line 115). You should replace “were placed” by “was placed” because the subject “amount” is singular.
Point 20 (Line 118). You should replace “were added” by “was added” because the subject “solution” is singular.
Point 21 (Figure 1). There is no trypsine in your scheme but it is mentioned in the text and pancreatin and amyloglucosidase are in the scheme but they are not described in the text. Explain it, please.
Point 22 (Line 128). You should replace “order” by “ordered”.
Point 23 (Line 135). You should insert the spaces before the abbreviations.
Point 24 (Line 142). You should replace your version by “All analyses were conducted in triplicate…” to make the sentence grammatically correct.
Point 25 (Line 144). You should omit the hyphen in “coefficient”.
Point 26 (Line 152). You should delete the italics in spelling “significant”.
Point 27 (Lines 162, 171). You should pay attention on the space in “Table_1” and “Table_2”.
Point 28 (Notes in the Table 1 and Table 2). You should write “coefficient of variation”.
Point 29 (Line 197). You should replace by “the findings show”.
Point 30 (Line 198). You should replace “need” by “needs” because the subject analysis is singular.
Point 31 (Line 200). You should omit the odd “between”.
Point 32 (Line 203). You should replace “given” by “having given” because it should be Perfect Participle 1 instead of Participle 2 “given” in order to express the extension action to predicate “performed” to make the sentence grammatically correct.
Point 33 (Line 206). You should replace “were subjected” by “was subjected” because the subject “ GPR” is singular.
Point 34 (Line 213). You should omit “are” after “parameters”.
Point 35 (Line 243). You should replace “is expressed” by “are expressed” because the subject “variations” is plural.
Point 36 (Line 250). You should insert the space before references.
Point 37 (Line 250). You should revise the spelling “determining”.
Point 38 (Line 251). You should insert “such as…” after “AC” and replace “are different” by “is different” because the subject “GPR” is singular.
Point 39 (Line 255). You should replace “found” by Past Perfect “had found” to make the sentence grammatically correct.
Point 40 (Line 256). You should replace the adjective “particular” by adverb “particularly”.
Point 41 (Line 259). You should replace participle “predicted” by noun “prediction”.
Point 42 (Line 262). You should check the spelling “formulae”.
Point 43 (Line 272). You should replace “diabetes” by “diabetics”.
Point 44 (Line 274). You should insert the space in “5._Conclusion”.
Round 2
Reviewer 1 Report
Dear authors,
The changes made in the paper have improved the quality and value of the paper.